# dCas9-Based PDGFR–β Activation ADSCs Accelerate Wound Healing in Diabetic Mice through Angiogenesis and ECM Remodeling

**DOI:** 10.3390/ijms24065949

**Published:** 2023-03-21

**Authors:** Yumeng Li, Deyong Li, Lu You, Tian Deng, Qiuyu Pang, Xiangmin Meng, Bingmei Zhu

**Affiliations:** Regenerative Medicine Research Center, West China Hospital, Sichuan University, Chengdu 610041, China

**Keywords:** chronic wounds, CRISPRa, PDGFR–β, ADSCs, extracellular matrix, angiogenesis

## Abstract

The chronic wound represents a serious disease characterized by a failure to heal damaged skin and surrounding soft tissue. Mesenchymal stem cells (MSCs) derived from adipose tissue (ADSCs) are a promising therapeutic strategy, but their heterogeneity may result in varying or insufficient therapeutic capabilities. In this study, we discovered that all ADSCs populations expressed platelet–derived growth factor receptor β (PDGFR–β), while the expression level decreased dynamically with passages. Thus, using a CRISPRa–based system, we endogenously overexpressed PDGFR–β in ADSCs. Moreover, a series of in vivo and in vitro experiments were conducted to determine the functional changes in PDGFR–β activation ADSCs (AC–ADSCs) and to investigate the underlying mechanisms. With the activation of PDGFR–β, AC–ADSCs exhibited enhanced migration, survival, and paracrine capacity relative to control ADSCs (CON–ADSCs). In addition, the secretion components of AC–ADSCs contained more pro–angiogenic factors and extracellular matrix–associated molecules, which promoted the function of endothelial cells (ECs) in vitro. Additionally, in in vivo transplantation experiments, the AC–ADSCs transplantation group demonstrated improved wound healing rates, stronger collagen deposition, and angiogenesis. Consequently, our findings revealed that PDGFR–β overexpression enhanced the migration, survival, and paracrine capacity of ADSCs and improved therapeutic effects after transplantation to diabetic mice.

## 1. Introduction

Chronic wound is a serious disease characterized by a failure to heal damaged skin and surrounding soft tissue. For instance, it is estimated that approximately 69.45 million people suffer from diabetic foot ulcers [1,2]. Notably, chronic wounds are among the leading causes of non–traumatic limb amputations associated with high morbidity and mortality [3]. Chronic wound healing is a complex biological process that involves the spatiotemporal coordination of multiple cell types and signaling pathways [4]. Certain pathological conditions, such as diabetes, inhibit normal wound healing, resulting in the formation of chronic, challenging-to-heal wounds. Moreover, the aging population has led to an increase in the number of patients with impaired wound healing [5]. Consequently, there is a growing need for novel therapeutic approaches to address this grave public health issue.

Due to the strong self-renewal, multi-directional differentiation and paracrine capacity, mesenchymal stem cells (MSCs) therapy may be an effective therapeutic option for chronic wounds [6]. It has been reported that MSCs therapy can promote the recovery process of diabetic wounds and other diseases [7,8]. Compared to other types of MSCs, MSCs derived from adipose tissue (ADSCs) have distinct advantages and fewer limitations: they are abundant, easy to harvest, and can be delivered through minimally invasive surgery, with fewer ethical issues, less risk of neoplastic and host immune responses [9,10]. Numerous studies have confirmed the therapeutic effect of ADSCs on wound healing [11,12,13,14]. However, the repair function of ADSCs is intrinsically constrained by the unfavorable microenvironment of wound sites [15,16]. Hence, ADSCs cannot play to their full therapeutic potential, preventing complete wound repair and regeneration [17]. Genetic engineering may endow MSCs with better abilities to fulfill more therapeutic potentials [18].

PDGFR–β, also known as CD140b, is widely expressed in perivascular mesenchymal cells, including pericytes and endothelial progenitor cells. Previous research has demonstrated that endogenous PDGFR–β signaling is essential for the formation of vascular wall cells. The absence of PDGFR–β impairs the proliferation and migration of vascular smooth muscle cells and pericytes in embryonic angiogenesis in mice [19]. Furthermore, it has been reported that PDGFR–β has a significant impact on fibroblasts during wound healing [20]. However, little is known about the effect of PDGFR–β on the therapeutic potential of ADSCs. In this study, using the SAM (synergistic activation mediator) system based on CRISPR–Cas9, we activated the expression of endogenous PDGFR–β in ADSCs (AC–ADSCs) and then assessed their proliferation, migration, secretion, and other functions in vitro. Following transplantation into diabetic mice, we evaluated their efficacy in promoting wound healing.

## 2. Results

### 2.1. Characterization of ADSCs and the Dynamic Expression of PDGFR–β in ADSCs

According to the definition of mesenchymal stem cells provided by the International Association for cell therapy, MSCs must exhibit adhesion, the expression of specific surface markers, and the capacity for three–dimensional differentiation (adipogenesis, osteogenesis and chondrogenesis). Five positive markers, CD105, CD90, CD73, CD44, and CD29, and five negative markers, CD45, HLA–DR, CD34, CD14, and CD19, were used as criteria for the lineage markers of MSCs in this study. As depicted in Figure 1A, flow cytometry analysis revealed that approximately 99% of ADSCs expressed positive markers (CD105, CD90, etc.) but no negative markers, exhibiting typical immunophenotypic characteristics of MSCs. In the induced medium, they successfully differentiated into adipogenesis, osteogenesis, and chondrogenesis (Figure 1B). Therefore, we have successfully isolated and cultured ADSCs and demonstrated their multidirectional differentiation potential.

Next, we investigated the expression pattern of PDGFR–β in ADSCs. According to a previous study [21], several receptor proteins, including PDGFR–α, could not be stably expressed in MSCs. Using PDGFR–β KO ADSCs (Appendix A) as control, flow cytometric analysis revealed that P5 generation ADSCs could uniformly express PDGFR–β (Figure 1C). When we further analyzed the expression of PDGFR–β among different passages by Western blotting, we discovered that the expression of PDGFR–β decreased with passage of ADSCs (Figure 1D,E), indicating unstable expression of PDGFR–β in different MSC passages.

### 2.2. PDGFR–β Enhances Migration and Paracrine of ADSCs

The above results indicated that the expression of PDGFR–β decreased with passage of ADSCs. Is this related to the diminished therapeutic potential of later passages of ADSCs? Using a CRISPR–Cas9-based SAM system, we activated PDGFR–β endogenously in ADSCs to determine the answer. After co-transfecting ADSCs with two lentiviruses, fluorescence microscopy images revealed concurrent expression of mCherry and EGFP (Figure 2B). In addition, approximately 23.4% of ADSCs were double–positive for mCherry and EGFP in flow cytometry sorting of the entire population (Figure 2C). After determining via Western blotting that PDGFR–β was significantly overexpressed in ADSCs (Figure 2D,E), we analyzed the stem cell identity of AC–ADSCs (PDGFR–β activation ADSCs). With PDGFR–β overexpression, the mRNA levels of three pluripotency–related genes, Sox2, Oct4, and Nanog, did not change significantly (Appendix A). In addition, the AC–ADSCs group was successfully induced into adipogenesis, osteogenesis, and chondrogenesis (Appendix A), indicating that AC–ADSCs could maintain their stem cell identity following PDGFR–β modification.

According to the experimental design, functional experiments were then conducted to determine whether the overexpression of PDGFR–β could affect the fundamental phenotypes of ADSCs. We observed that after endogenous activation of PDGFR–β, there was no significant difference in the proliferation of ADSCs compared to the control group within three days (Figure 2G), whereas the migration of AC–ADSCs was increased as determined by the scratch test (Figure 2F). In addition, we discovered that after PDGFR–β overexpression, the protein expression of p–AKT/AKT, platelet derived growth factor subunit B (PDGFB), a precursor to PDGFBB that is closely associated with wound repair, was significantly increased (Figure 2H,I). The PI3K/AKT pathway is essential for radical cellular activities including survival and paracrine signaling [22,23]. Therefore, we further detected the paracrine and survival under stress of ADSCs following PDGFR–β overexpression. Using RT–qPCR and ELISA, the expression levels of several growth factors associated with wound healing and angiogenesis were analyzed. Results of RT–qPCR showed that in endogenously activated ADSCs, the transcription levels of *bFGF*, *ANG1*, *TGFβ*, *CXCL12*, *COL1a1*, *COL3a1*, and *VEGFa* were significantly higher than those in the control group (Figure 2J), and ELISA results demonstrated that in the AC–ADSCs, the secretion levels of IL–10, TGF–β1, VEGFA, and HGF was significantly higher than in the control group (Appendix A). Oxidative stress was induced by H_2_O_2_ treatment for 10 h in ADSCs to test the survival under stress. As STAT3 plays a central role for the regulation of cell growth, differentiation, and survival, and can be activated via tyrosine kinase receptors [24], we detected p–STAT3 expression through Western blotting. The results demonstrated that the expression of p–STAT3/STAT3 was significantly elevated in AC–ADSCs, regardless of whether they were stimulated with H_2_O_2_. In addition, as the downstream target of STAT3, the expression of BCL–2 was significantly increased in PDGFR–β overexpressed ADSCs (Figure 3A,B); and flow cytometry analysis revealed that less apoptosis was detected in the AC–ADSCs, compared to the CON–ADSCs, indicating improved viability under H_2_O_2_ treatment (Figure 3C,D). The data presented in this section demonstrated that PDGFR–β overexpressed ADSCs exhibit significantly enhanced migration, anti–oxidative stress capacity, and paracrine activity compared to the control group, while retaining stem cell identity.

### 2.3. PDGFR–β–ADSCs Conditioned Medium Promotes the Function of HUVEC In Vitro

In chronic wound healing, impaired angiogenesis is a major contributor to delayed wound healing. MSCs can induce angiogenesis via paracrine activation of endothelial cells; consequently, we examined the cell-to-cell interactions between MSCs and endothelial cells. In accordance with the design of the in vitro crosstalk experiments, conditioned medium from AC–ADSCs and CON–ADSCs cultured for 48 h was collected and used for HUVECs.

The scratch experiment demonstrated that the conditioned medium of ADSCs could significantly enhance the migration of HUVECs compared to the basal medium (α–MEM), while the AC–CM treatment group demonstrated the most effective wound closure among the three groups (Figure 4A,B). Additionally, CCK8 analysis demonstrated that the conditioned medium of ADSCs could significantly enhance the proliferation of HUVECs compared to the basal medium group at all time points, and the HUVECs treated with the AC–CM exhibited the highest cell viability of the three groups (Figure 4C). In addition, tubule formation in ECs exposed to ADSC–CM was evaluated. As shown in Figure 4D–F, AC–CM significantly improved the number of junctions and number of meshes in comparison to the other two groups, while CON–CM also outperformed the α–MEM group. Besides, the mobilization of AKT and ERK signaling is essential for the functional performance of HUVECs. Therefore, we further investigated the activation of AKT and ERK signaling in HUVECs after 48 h of conditioned culture. Among the three groups, AC–CM treatment activated the ERK and AKT signals in HUVECs to the greatest extent, strengthening the function of ECs in the wound microenvironment (Figure 4G,H).

### 2.4. PDGFR–β Overexpressed ADSCs Conditioned Medium Promoted the Expression of ECM Related Genes in HaCaT Cells

Wound healing is a complex and orderly process that involves many different types of cells. Keratinocytes are primarily responsible for rapid wound closure and structural repair. Therefore, we evaluated the effect of ADSC–CM on keratinocytes through in vitro functional experiments.

Intriguingly, ADSCs–CM was able to significantly increase the migration and proliferation of HaCaT cells in the scratch test and CCK8 test when compared to the α–MEM group, whereas the AC–CM treatment group did not differ significantly from the CON–CM treatment group (Figure 5A–C). Therefore, we examined the expression levels of genes involved in ECM remodeling in HACAT cells 48 h after treatment with CON–CM or ADSCs–CM. In comparison to the CON–CM, the AC–CM significantly increased the transcription levels of *PDGFB*, laminin (*LN*), *TGF–β*, and fibronectin (*FN*) in HaCaT cells as determined by RT–qPCR (Figure 5D).

### 2.5. PDGFR–β Overexpressed ADSCs Accelerated Wound Healing in Diabetic Mice via ECM Remodeling and Angiogenesis

To evaluate the effect of endogenously PDGFR–β activated ADSCs on chronic wound healing in vivo, we subcutaneously transplanted ADSCs into diabetic mice at four points around the wound site at 7.5 × 10^5^ per wound in 100 uL PBS, and continuously monitored wound closure (Figure 6A).

The ADSCs treatment groups demonstrated significantly faster wound healing than the PBS group three days after transplantation. Within the two ADSCs treatment groups, AC–ADSCs treatment was significantly superior to CON–ADSCs treatment beginning on the third day (Figure 6B,C). On day 9, we collected additional samples and utilized western blotting to analyze the expression levels of several proteins associated with wound healing and angiogenesis. The results demonstrated that the levels of VEGFA and PDGFR–β in ADSCs treatment groups were significantly higher than those in the PBS treatment group, whereas the levels of VEGF–A, PDGFB, PDGFR–β, p–AKT/AKT, and p–STAT3/STAT3 in AC–ADSCs group were significantly higher than those in CON–ADSCs group (Figure 6D,E). The data presented above suggested that ADSCs could promote wound healing in diabetic mice, and the overexpression of PDGFR–β in ADSCs could further accelerate the healing process in vivo.

To further elucidate how AC–ADSCs promoted chronic wound healing in diabetic mice, we analyzed cell residency, ECM remodeling, and angiogenesis based on our previous findings. And tissue samples were collected for pathological analysis on day 3 and 9.

After transplantation of DiR-labelled CON–ADSCs and AC–ADSCs, in vivo imaging system (IVIS) photographs of mice with dorsal wounds were taken on D0 and D5, and the results demonstrated that within five days after MSCs transplantation, the AC–ADSCs transplantation group had a lower signal attenuation rate than the CON–ADSCs transplantation group, indicating their higher survival rate and residence rate in the wound microenvironment than CON–ADSCs (Figure 7A).

Besides, the AC–ADSCs treatment group exhibited a more rapid response in the early phase of wound healing, as indicated by higher expression of COL3a1 around wounds and extracellular matrix-related genes, including *col3a1*, *α–sma*, *col1a1*, and *tgf–β* (Figure 7B,C). In the middle and late stages of wound healing, the cross–section of the AC–ADSCs treatment group was significantly smaller than that of the CON–ADSCs treatment group and the PBS treatment group, as determined by Masson staining (Figure 7D,E). In addition, immunofluorescence staining confirmed that the positive signals of CD31 and α–SMA were significantly higher in the AC–ADSCs group than in the CON–ADSCs group and the PBS group around the wound (Figure 7F–H), indicating that stronger angiogenesis developed in chronic wounds after AC–ADSC transplantation.

### 2.6. Proteomics of CON–ADSCs CM and AC–ADSCs CM

Collectively, our findings indicated that AC–ADSC may facilitate wound healing via promoted secretory components. Moreover, the proteomics of CON–CM and AC–CM were conducted to explore the changes in ADSCs secretory components caused by PDGFR–β modification from a global view. The proteomics experiment identified a total of 3608 proteins. The GO analysis classified the outcomes according to biological process, cellular component, and molecular function. The identified proteins in the cellular component were significantly enriched in the extracellular and cytoplasmic regions (Figure 8A). Regarding biological processes, ADSCs–CM proteins were predominantly involved with cellular component, cytoplasmic translation, and protein–containing complex-related processes. As for molecular function, proteins were abundant in nucleic acid and protein binding (Figure 8A).

Following the identification of differentially expressed proteins, a functional clustering analysis was performed. Hence, 943 proteins were identified with Log2|FC| ≥ 2, *p*-value < 0.05 as the screening standard (Figure 8B), and upregulated proteins in AC–CM were selected for functional cluster analysis. The heat maps revealed that endogenous activation of PDGFR–β led to a significant upregulation of ADSCs in biological processes such as angiogenesis, wound healing, and extracellular matrix remodeling (Figure 8C–E). The cluster analysis of the KEGG database revealed that the PI3K–AKT pathway and the ECM–related pathway were upregulated (Figure 8F), which is consistent with our in vitro and in vivo findings (Figure 4G and Figure 6D).

## 3. Discussion

Numerous receptor proteins, including PDGFR–α, were not expressed uniformly throughout the entire MSC population, which may lead to diminished therapeutic potential according to previous research [21,25]. In this study, we discovered that PDGFR–β was expressed uniformly in all MSC populations, whereas its expression decreased with passage. Given that this was associated with decreased therapeutic efficacy of MSCs in later passages and that PDGFR–β was essential for MSC therapy, PDGFR–β overexpression may enhance the therapeutic potential of ADSCs. The CRISPRa system was used to overexpress PDGFR–β in ADSCs, resulting in improved migration, anti–apoptosis, and paracrine, without affecting stemness. Besides, chronic wound healing in mice was accelerated by AC–ADSCs due to enhanced angiogenesis and extracellular matrix remodeling.

In the CRISPRa system, point mutations in Cas9 (D10A and H840A) result in a deactivated form of Cas9–dCas9, which is fused with transactivation domains such as VP64 and P300 to precisely activate gene expression. This has been utilized as a potent tool for the endogenous overexpression of large genes. Jianglin Wang et al. reprogrammed fibroblasts into induced cardiac progenitor cells using CRISPRa–based transcriptional activators [26], and Chih–Hao Wang et al. engineered human white preadipocytes to produce human brown–like (HUMBLE) cells by activating endogenous uncoupling protein 1 expression with CRISPR–Cas9–SAM–gRNA [27]. However, no CRISPRa–based therapeutic approaches have been reported, and numerous researchers have utilized conventional technologies to implement cell therapies. Robert H. Baloh et al. demonstrated in clinical trials that human neural progenitor cells transduced with GDNF (CNS10–NPC–GDNF) via exogenous transduction of lentivirus differentiated into astrocytes and were safe for 18 ALS patients in a phase 1/2a study (NCT0294388) [28]. Von Einem et al. obtained MSC–apceth–101 through gamma-retroviral transduction and utilized it to treat patients with advanced gastrointestinal adenocarcinoma [29]. Francesca Fumagalli et al. transduced the ARSA gene into the autologous hematopoietic stem cells of MLD patients using lentiviral exogenous transduction [30]. An increasing number of experimental as well as clinical studies have shown the promise of gene therapy and genetically engineered cell therapy. Genetically engineered cells do increase the efficacy of treatment, and gene–modified cell therapy appears to be safe and feasible at same time. In this study, we found that all ADSCs populations could continuously express PDGFR–β while the expression level decreased with passage, which maybe the cause of insufficient therapeutic potential reported in some clinical trials [31,32,33]. Moreover, if the PDGFR–β of MSCs used in these clinical trials were activated by the CRISPRa, it is possible that better therapeutic effects would be presented. In addition, similar to hypoxic preconditioning, gene editing can be viewed as a pretreatment prior to the application of mesenchymal stem cells to improve the specific functions of MSCs and achieve artificial standardization against the unstable effects caused by the heterogeneity of mesenchymal stem cells. Moreover, besides PDGFR–β, there are many other valuable therapeutic molecules whose size is too large to be packaged in the virus. For these molecules, CRISPRa is really a powerful tool to achieve gene editing in cell therapy. We concluded that the CRISPRa system shed light on endogenous gene activation for cell therapies.

The tyrosine kinase receptor PDGFR–β (CD140b) is essential for angiogenesis and early hematopoiesis. It is expressed in smooth muscle cells and pericytes of the vascular system [19,34]. The PDGFR–β signaling pathway is essential for the recruitment and differentiation of vascular smooth muscle/pericyte progenitor cells during vascular development [19,34]. In addition, PDGFR–β is essential for pericyte recruitment, proliferation, and functional activity during wound healing and remodeling [35]. Similarly, knockout of PDGFR–β in mouse fibroblasts has been reported to result in delayed wound healing [20]. Numerous previous studies have demonstrated the importance of endogenous PDGFR–β signaling in vascular development and tissue repair, whereas the function of PDGFR–β signaling in exogenous transplanted therapeutic cells remains poorly understood. Andrew Owen et al. reported that PDGFR–α positive MSCs had a better therapeutic effect on a rat model of hepatic ischemia-reperfusion [36], and Wang Feng et al. demonstrated that PDGFR–α positive BMSCs can secrete more mir–6924–5p, which can better treat osteolysis and improve healing intensity during tendon bone healing [37]. Consequently, as a member of the PDGFRs family, what effect does PDGFR–β signaling have on the function and therapeutic potential of ADSCs? Contradicting some previous reports that some growth factor receptors could not be expressed continuously in all MSC populations [21], PDGFR–β is expressed continuously in all ADSCs in this study. Also, we found that the alteration in PDGFR–β signaling caused by gene editing with a CRISPR–SAM system did not affect the stemness of ADSCs, which is different from previous researches [38,39]. Jiezhong Chen et al. demonstrated that PI3K/Akt signaling is essential for MSCs’ survival, proliferation, migration, angiogenesis, cytokine production, and differentiation [40]. In this study, we identified the activation of AKT signaling following the overexpression of PDGFR–β, and our in vivo and in vitro data demonstrated that the migration, paracrine, and anti–apoptotic effects of ADSCs were significantly enhanced by the activation of AKT signaling. In addition, we observed that the PDGFR–β signaling mobilized p–STAT3, which is also a vital survival–related signal.

Accumulating evidence suggested that MSC therapy promoted repair through paracrine mechanisms [41,42,43]. Accordingly, we examined the cell–to–cell cross talk in vitro using the conditioned medium of ADSCs.

In the initial stages of wound healing, keratinocytes interact with fibroblasts to initiate epithelial–to–mesenchymal transition in order to promote rapid wound closure and ECM homeostasis [44]. Intriguingly, our data indicated that AC–CM did not enhance the proliferation and migration of HaCaT cells in vitro relative to CON–CM, whereas it did stimulate the expression of ECM genes. Moreover, in vivo experiments conducted during the initial phase of wound healing demonstrated that transplantation of AC–ADSCs promoted collagen deposition and wound closure. These results suggested that perhaps the overexpression of PDGFR–β prevents ADSCs from secreting sufficient substances to mobilize HaCaT cells, but that they can directly secrete more substances of ECM composition, as supported by our proteomic clustering results.

Angiogenesis is an additional crucial wound healing process. Under normal conditions, angiogenesis primarily occurs during the proliferative phase of wound healing, and these newly formed small blood vessels are essential for supplying nutrients and oxygen to the skin for regeneration [45]. Nonetheless, abnormal patterns of growth factor secretion significantly impair angiogenesis in chronic wounds, leading to the formation of defective granulation tissue that impedes the progression of wound healing to the proliferative phase [46]. Overexpression of PDGFR–β in ADSCs improved the ability of AC–ADSCs to promote the function of ECs in vitro via secretion components, resulting in enhanced angiogenesis in mice with chronic wounds. This may be due to the increased pro–angiogenic and ECM–related components secreted by AC–ADSCs as a result of the AKT signaling, or it may be due to the increased residency and survival of engineered ADSCs.

In addition to the new findings revealed by this research, there were also some limitations. First, we primarily concentrated on the functional and phenotypic enhancement of engineered MSCs, while mechanistic research was lacking. Second, MSC–EV is gaining popularity as a novel cell derivative therapy and demonstrating therapeutic potential for a variety of diseases [47,48,49]. We demonstrated that PDGFR–β signaling could enhance the paracrine activity of ADSCs, but we did not investigate the changes in EV secretion. In our follow–up research, we should investigate the EV secretion profile of ADSCs with and without PDGFR–β overexpression.

## 4. Materials and Methods

### 4.1. Isolation of ADSCs

The ADSCs were derived from human adipose tissue, isolated and minced using a blade, and then digested with 0.1% collagenase type I (Gibco Life Technologies, New York, NY, USA) for 2–3 h at 37 °C. The digestion was terminated by Minimum Essential Medium–α (α–MEM) supplemented with 5% Serum substitute (Helios Bioscience, HPCFDCRL50, Atlanta, GA, USA). The solution was then passed through a 100–μm filter and centrifuged at 1400 rpm for 5 min. Subsequently, the supernatant was discarded, and the deposit washed with α–MEM twice, then resuspended and cultured in a T75 flask.

### 4.2. Phenotypic Analysis and PDGFR–β Expression by Flow Cytometry

After washing twice with PBS, cells were resuspended and incubated with pre–labelled antibodies for 15 min at room temperature. After two washes with PBS, cells were resuspended in 300 µL PBS and analyzed using a flow cytometer (Beckman Coulter Life Sciences, Indianapolis, IN, USA). Histograms were generated using the cytexpert and FlowJo software. The antibodies used were as follows: FITC anti–human CD34 (Abcam, ab195013, Waltham, MA, USA), FITC anti–human CD45, (BD Biosciences, 557803, Piscataway, NJ, USA), PE–Cy7 anti–human CD14 (BD Biosciences, 561385, Piscataway, NJ, USA), PERCP–CY5.5 anti–human HLA–DR (BD Biosciences, 552764, Piscataway, NJ, USA), PERCP–CY5.5 anti–human CD73 (BD Biosciences, 561260, Piscataway, NJ, USA), PE anti–human CD90 (BD Biosciences, 555596, Piscataway, NJ, USA), APC anti–human CD19 (eBioscience, 11–0199042, San Diego, CA, USA), PE anti–CD105 (eBioscience, 25–1057–42, San Diego, CA, USA), PE anti–human CD44 (BD Biosciences, 555479, Piscataway, NJ, USA), APC anti–human CD29 (BD Biosciences, 559883, Piscataway, NJ, USA), and BV421 anti–human PDGFR–β (BD Biosciences, 564124, Piscataway, NJ, USA).

### 4.3. Cell Differentiation Capacity Analysis

For the analysis of adipogenic differentiation capacity, 5 × 10^4^ cells were plated in six–well plates. When reaching 100% confluence, the medium was changed to the adipogenic differentiation basal medium–A (ADBM–A, Cyagen, Santa Clara, CA, USA) for 3 d. Subsequently, the ADBM–A was changed to the ADBM–B (Cyagen, Santa Clara, CA, USA) for 1 d. ADBM–B maintained cells until the lipid droplets became larger after ADBM–A and ADBM–B were used alternately for 3 to 5 times. Finally, hADSCs were fixed in 4% paraformaldehyde for 10 min, stained with fresh Oil Red–O (Sigma–Aldrich, 1320-06-5, New Brunswick, NJ, USA) solution to stain lipid droplets, and then photographed. For the detection of osteogenic differentiation capacity, 5 × 10^4^ cells were plated in six–well plates until 70% confluence. The cells were then cultured in osteogenic differentiation basal medium (ODBM, Cyagen, Santa Clara, CA, USA) for 3 weeks before being fixed in 4% paraformaldehyde and stained by Alizarin Red. For the study of chondrogenic differentiation capacity, 5 × 10^5^ cells were centrifuged and collected in the bottom of 15 mL centrifuge tube. The culture medium was changed to the chondrogenic differentiation basal medium (CDBM, Cyagen, Santa Clara, CA, USA) for 4 weeks after cells were gathered into pellets. Alcian blue staining was used to evaluate the capacity of hADSC pellets to differentiate toward chondrocyte pellets.

### 4.4. Ex Vivo Genome Editing of ADSCs

Human ADSCs were cultured to 80–90% confluency before being trypsinized (Thermo Fisher, A1217703, Waltham, MA, USA) and resuspended to 5000 cells/µL in Opti–MEM^®^ I (Invitrogen, 11058021, New York, NY, USA). Nucleases and sgRNA precomplexed RNP (10min at room temperature, 2.5:1 molar sgRNA–to–Cas9 ratio, final concentration: 0.3 µg/µL Cas9 protein (Integrated DNA Technologies, 1081059, Coralville, IA, USA). Then, hMSC–Nuclease mixtures were electroporated with pulse code CM–119 on the Lonza 4D Nucleofector (Lonza, Basel, Switzerland) and immediately diluted with 2 × volume Opti–MEM^®^ I, followed by inoculation and culture.

### 4.5. Lentivirus Transduction and FACS

Cells were plated at 3000 cells/cm^2^ for one day and then co–transduced with two lentiviruses encoding, sgRNA–MS2–HSF1–P65–mcherry and dcas9–EGFP, separately. After eight hours, the cells were washed with PBS and replaced with fresh complete medium. Then, cells were sorted by flow cytometry to obtain EGFP and mCherry double–positive ADSCs for subsequent experiments 72 h post transduction.

### 4.6. Flow Cytometry and FACS

GFP+ and mCherry^+^ cells were sorted by flow cytometry 3 days after virus transfection and expanded for further analysis.

### 4.7. Real Time Quantitative Reverse Transcription Polymerase Chain Reaction (RT–qPCR)

RNA was extracted from control ADSCs (CON–ADSCs) and PDGFR–β activation ADSCs (AC–ADSCs) using TRIZOL reagent (Invitrogen™, 15596026, New York, NY, USA). Total RNA (1.0 ug) was reversely transcribed into cDNA using Hiscript III Reverse Transcriptase (Vazyme, R302, Nanjing, China). qPCR was conducted using a Hieff® qPCR SYBR Green Master Mix (Yeasen, 11201ES50, Shanghai, China) and an ABI 7900HT fast real-time PCR system (Applied Biosystems, QuantStudio 6 Flex, Ajo, AZ). Each sample was measured three times. Detailed primer sequences are shown in Appendix A, including VEGF, HGF, IL–10, ANG1, CXCL12, COL1, COL3, bFGF, and TGF–β. All genes were normalized to the endogenous reference gene ACTB. mRNA expression levels of target genes were calculated by the DDCt method.

### 4.8. Conditioned Medium of ADSCs

CON–ADSCs and AC–ADSCs were cultured in 10 cm dishes. After growing to 70–80% confluence, following washed with PBS, the medium was replaced with basal medium (α–MEM). After that, the cells were cultured for 48 H for the collection of conditioned medium. The conditioned medium was centrifuged at 3000 rpm for 5 min at 4°C and stored at −80 °C.

### 4.9. ELISA Analysis

The conditioned medium (CM) of ADSCs was collected as previously described. An ELISA assay was performed to evaluate secretion levels of IL–10 (Elabscience, E–EL–H6154), TGF–β (Elabscience, E–EL–0162c), VEGFA (Elabscience, E–EL–H0111c), and IGF (Elabscience, E–EL–H0086c) in the CM according to the manufacturer’s instructions.

### 4.10. Western Blot Analysis

Skin tissues and cell suspension were lysed by RIPA lysis buffer (MB–030–0050, Multi Sciences Biotech, Hangzhou, China) complemented with phenylmethylsulfonyl fluoride. A BCA Protein Assay Kit (23225, Thermo Fisher, Waltham, MA, USA) was adopted to determine protein concentrations. After establishing equal quantities, total proteins were separated in SDS–PAGE gel (4–20%, ACE Biotechnology, Nanjing, China), and the bands were transferred onto PVDF membranes (Millipore, NJ, USA). After blocking with 5% skimmed milk and washed with tris–buffered saline containing 0.1% tween–20 (TBST), protein bands were blotted with primary antibodies at 4 °C overnight (Antibody information is provided in Appendix A). After washing with TBST, protein bands were blotted with HRP conjugated secondary antibody and monitored using ECL buffer (32209, ThermoFisher, Waltham, MA, USA). Quantifications of western blotting were measured with ImageJ.

### 4.11. Diabetic Wound Healing Evaluation

Male C57BL/6 mice aged 8–10 weeks were purchased from GemPharmatech Co. Ltd., (Chengdu, China), and raised at the specific pathogen–free laboratory animal facility of West China Hospital, Sichuan University (Chengdu, China). Mice were intraperitoneally injected with streptozotocin (120 mg/kg in 0.1 M citrate–buffered saline, pH 4.5, Sigma, S0130) to induce Diabetes Mellitus (DM). Glucose was measured with blood sugar test paper (Sinocare, 6243578, Changsha, China). Glucose levels >16.7 mM were diagnosed with diabetes. To establish a stable animal model of DM, diabetic mice were fed for 4 weeks with normal diet, and blood glucose levels were reconfirmed before wound formation. After anesthetized by pentobarbital sodium (i.p. injection, 50 mg/kg), a square full thick skin injury (4 mmin diameter) was produced on the back of each diabetic mice and silicone rings were sutured around the wounds to prevent contracture. Then, diabetic mice were randomly divided into three groups, respectively, in which CON–ADSCs, AC–ADSCs (7.5 × 10^5^ cells in 100 μL PBS), or 100 μL PBS were subcutaneously injected into four points around the wound edge. The wounds were photographed on days 0, 3, 5, 7, and 9, respectively after surgery to observe the healing process. Image J software was used to analyze wound size on days 0, 3, 5, 7, and 9, respectively.

### 4.12. Cell Labeling with DIR

Injected cells were labeled with the fluorescent tracer 1, 1–dioctadecyl–3,3,3,3–tetramethyl in dotricarbocyanine iodide (DiR; Caliper Life Sciences, Hopkinton, MA, USA) following to the manufacturer’s protocol. Briefly, cells were incubated with DiR for 30 min at 37 °C, centrifuged for 5 min at 1500 rpm at room temperature, and then rinsed twice with PBS. In all the cases, DiR–labeled cells were suspended in PBS, and 7.5 × 10^5^ cells were transplanted into diabetic mice within 2 H after labeling.

### 4.13. IVIS Observation

DiR–labeled CON–ADSCs (*n* = 5) and AC–ADSCs (*n* = 5) were transplanted around the wound within 2 h of cutting at a dose of 7.5 × 10^5^/wound. Filter conditions and illuminations settings for DiR imaging were set an excitation/emission of 710/760 nm, high lamp level, medium binning, filter 1, and 1.0 sec exposure time. Grayscale and fluorescent images of each sample were analyzed using Living Image software version 4.3 (Xenogen). Regions of interest of each sample were automatically drawn over the signals on images and, if necessary, they were manually corrected according to the grayscale image. Quantification was made according to the method of Cho et al. with modification. The distribution of each DiR–labeled cell in each organ was quantified as the average radiant efficiency (total photons/s/cm^2^/steradian) in the irradiance range (μW/cm^2^): (photons/s/cm^2^/steradian)/(μW/cm^2^). To reduce variability in measurements, the ratio of the average radiant efficiency of the organs to the background was calculated.

### 4.14. Histological Observation

The tissue samples harvested on day 3 and 9 post wounding were fixed in 4% paraformaldehyde overnight at 4 °C. After being washed with PBS, they were dehydrated in a graded ethanol series (30%, 50%, 70%, 80%, 90%, and 100%), xylene, and paraffin washes and embedded in paraffin. Sections (4–6–mm–thick) were prepared from the paraffin–embedded wound tissues and then stained with Masson’s trichrome. Tissue sections were observed using a microscope.

### 4.15. Proteomic Analysis

CON–ADSCs and AC–ADSCs were seeded at 3 × 10^5^ cells mL^−1^, followed by treatment with serum free α–MEM for 48 h. Finally, after centrifugation (1200× *g*/10 min) to remove the sediment, conditioned medium (CM) were collected and label–free quantitative proteomics and bioinformatics analyses were performed by Bioprofile in Shanghai. Three independent tests were carried out for each group. The data were analyzed and plotted using R software. When selecting protein accumulation or reduction, log2(FC) ≥2 and *p* < 0.05 were chosen as the criteria. The GO term enrichment analysis was performed through the “cluster profiler” and “org.Hs.eg.db” R packages.

### 4.16. Statistical Analysis

Data from the mouse and cell model were expressed as mean ± SD. Significant differences were assessed either by Independent–sample t test (two–tailed) was used for statistical comparisons between 2 groups, or one–way ANOVA followed by Tukey post hoc test for statistical comparisons between multiple groups. A value of *p* < 0.05 was considered to be statistically different. Analyses were performed using GraphPad Prism.

## 5. Conclusions

Based on the presented results, it may be concluded that the sustained overexpression of PDGFR–β achieved through CIRSPR activation in ADSCs activated Akt signaling, which led to significant functional improvements in the migration, survival, and paracrine of ADSCs, thereby enhancing the therapeutic effect in mouse diabetic wounds.

## Figures and Tables

**Figure 1 ijms-24-05949-f001:**
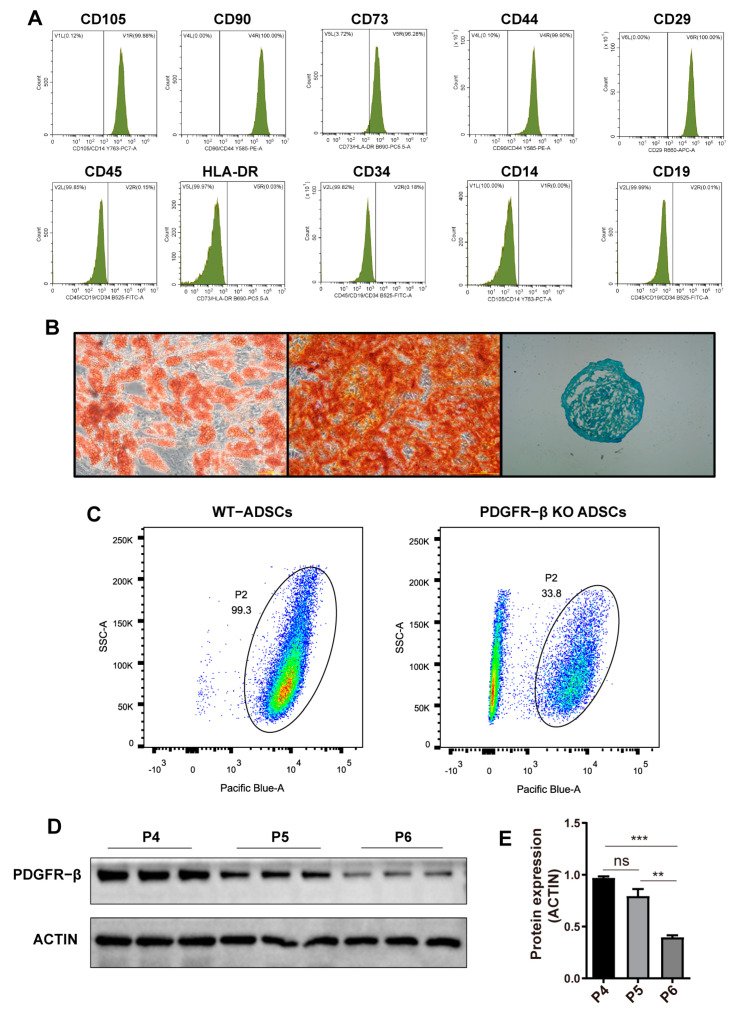
PDGFR–β is dynamically downregulated with ADSC passage. (**A**) Cell surface co–expression of the antigens, CD44, CD73, CD90, and CD105 in MSCs. (**B**) Differentiation potential of MSCs in osteogenic, chrondrogenic, and adipogenic lineages using Alizarin red, Alcian blue, and oil red O staining, respectively. (**C**) Flow cytometry analysis indicating the expression pattern of PDGFR–β in single cell suspensions of ADSCs (CD105–PE, CD90–PE, CD73–CY5.5, CD44-PE, CD29-APC, CD45-FITC, HLA-DR-CY5.5, CD34-FITC, CD14-PE, CD19-FITC). (**D**) PDGFR–β expression and (**E**) levels relative to β–ACTIN in P4–P6 ADSCs (*n* = 3). Data are shown as mean ± SD. Independent–sample *t* test (two–tailed) was used for statistical comparisons between 2 groups; One–way ANOVA followed by Tukey post hoc test was used for statistical comparisons between multiple groups. *** p* < 0.01; **** p* < 0.001.

**Figure 2 ijms-24-05949-f002:**
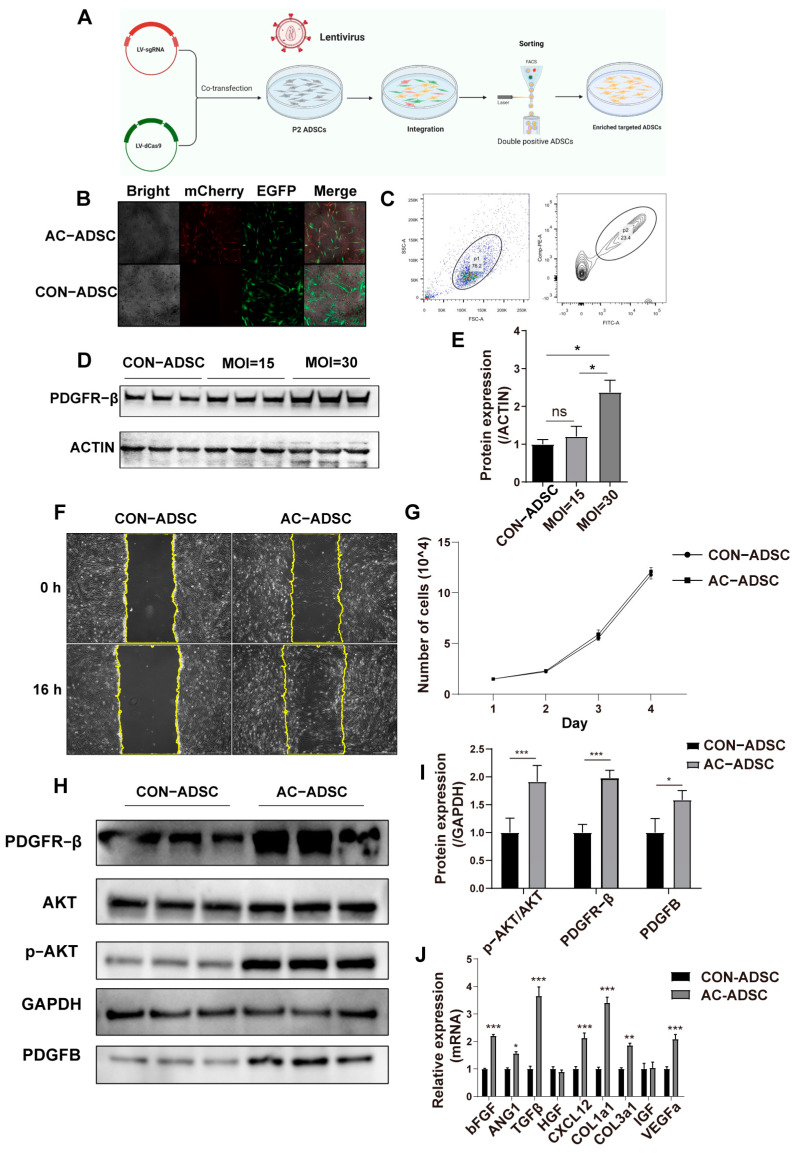
Endogenously activated PDGFR–β in ADSCs through SAM system. (**A**) Two-lentivirus system coinfection and sorting process. (**B**) Representative single-channel and overlay images of mCherry + and GFP + ADSCs. (**C**) Sorting of mCherry and GFP double positive ADSCs. (**D**) PDGFR–β expression and (**E**) levels relative to ACTIN in ADSCs with different multiplicity of infection (*n* = 3). (**F**) In vitro migration of CON–ADSCs and AC–ADSCs determined by scratch test. (*n* = 6, scale bar = 200 μm). (**G**) Growth curve of AC–ADSC and CON–ADSC (*n* = 3). (**H**,**I**) Western blotting analysis of the levels of PDGFR–β, PDGFB, p–AKT in CON–ADSCs and AC–ADSCs (*n* = 3). (**J**) Expression of some growth factors and cytokines detected by RT–qPCR (*n* = 3). Data are shown as mean ± SD. Independent-sample *t* test (two-tailed) was used for statistical comparisons between 2 groups; One–way ANOVA followed by Tukey post hoc test was used for statistical comparisons between multiple groups. ** p* < 0.05; *** p* < 0.01; **** p* < 0.001.

**Figure 3 ijms-24-05949-f003:**
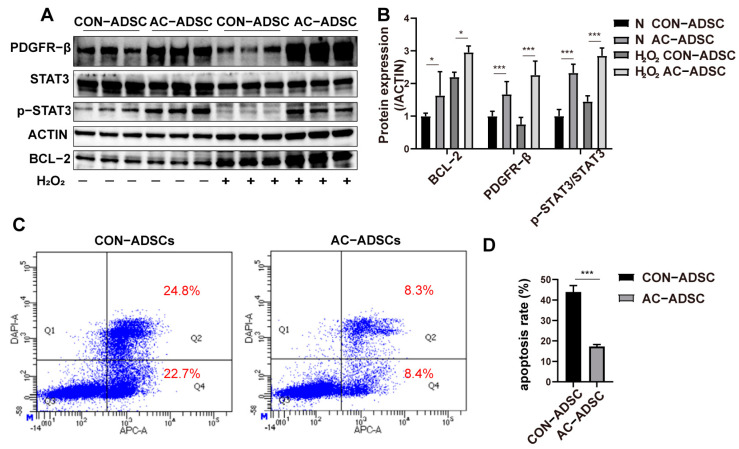
PDGFR–β overexpression enhances the resistance to oxidative stress of ADSCs. (**A**) Western blotting and (**B**) quantification analysis of H_2_O_2_ treated ADSCs (*n* = 3). (**C**) The apoptosis of ADSCs detected by flow cytometry and (**D**) quantification analysis (*n* = 3). Data are shown as mean ± SD. Independent–sample *t* test (two–tailed) was used for statistical comparisons between 2 groups; One–way ANOVA followed by Tukey post hoc test was used for statistical comparisons between multiple groups. ** p* < 0.05; *** *p* < 0.001.

**Figure 4 ijms-24-05949-f004:**
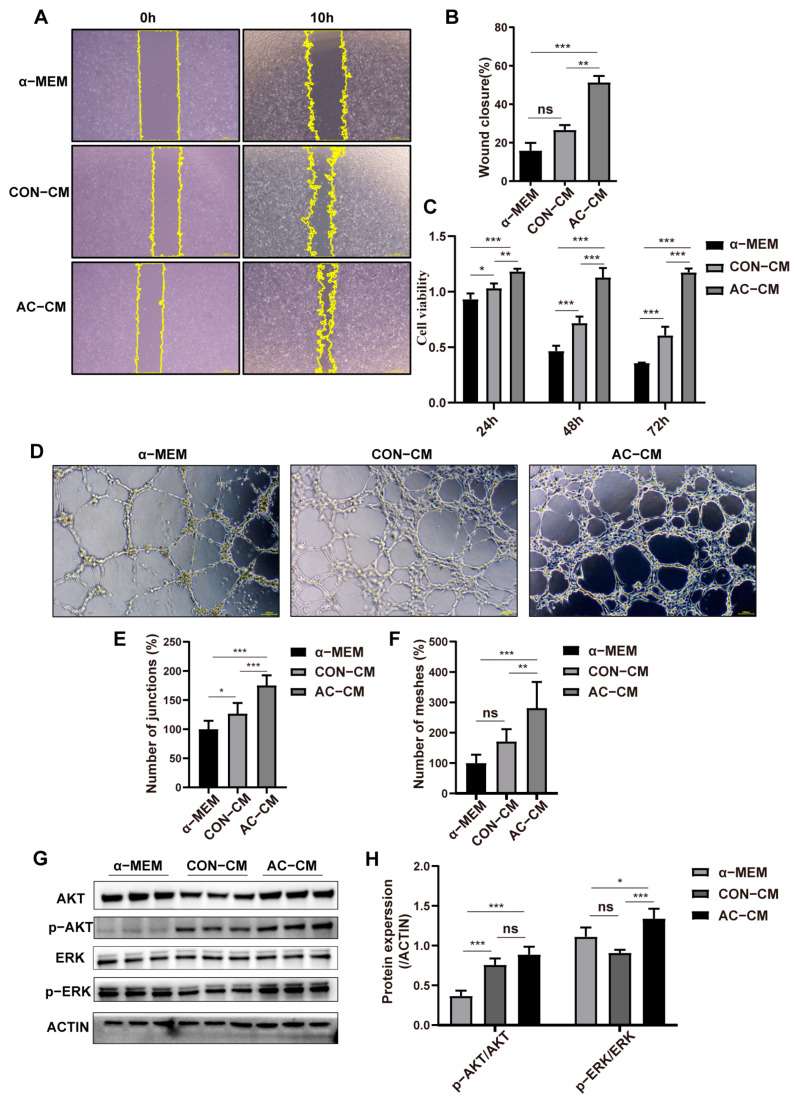
Conditioned medium from AC–ADSCs promotes endothelial function. (**A**) Representative scratch test image of HUVECs incubated with conditioned medium of CON–ADSCs and AC–ADSCs at 0 and 10 h, and (**B**) Rate of wound area closure after 10 h (%) (*n* = 6, scale bar = 100 μm). (**C**) CCK8 assay was used to evaluate the viability of HUVECs incubated with conditioned medium of CON–ADSCs and AC–ADSCs (*n* = 6), and the statistical analysis of HUVECs viability within 72 h (%). (**D**) Representative tube formation images of HUVECs incubated with conditioned medium of CON–ADSCs and AC–ADSCs for 8 h (*n* = 3, scale bar = 100 μm), and (**E**) quantitative analysis of number of junctions, (**F**) number of meshes and per field. (**G**) Representative images of western blotting and (**H**) quantification analysis of ERK and AKT signaling pathways proteins expression in HUVECs incubated with conditioned medium of CON–ADSCs and AC–ADSCs (*n* = 3). Data are shown as mean ± SD. Independent–sample *t* test (two-tailed) was used for statistical comparisons between 2 groups; One-way ANOVA followed by Tukey post hoc test was used for statistical comparisons between multiple groups. ** p* < 0.05; *** p* < 0.01; **** p* < 0.001.

**Figure 5 ijms-24-05949-f005:**
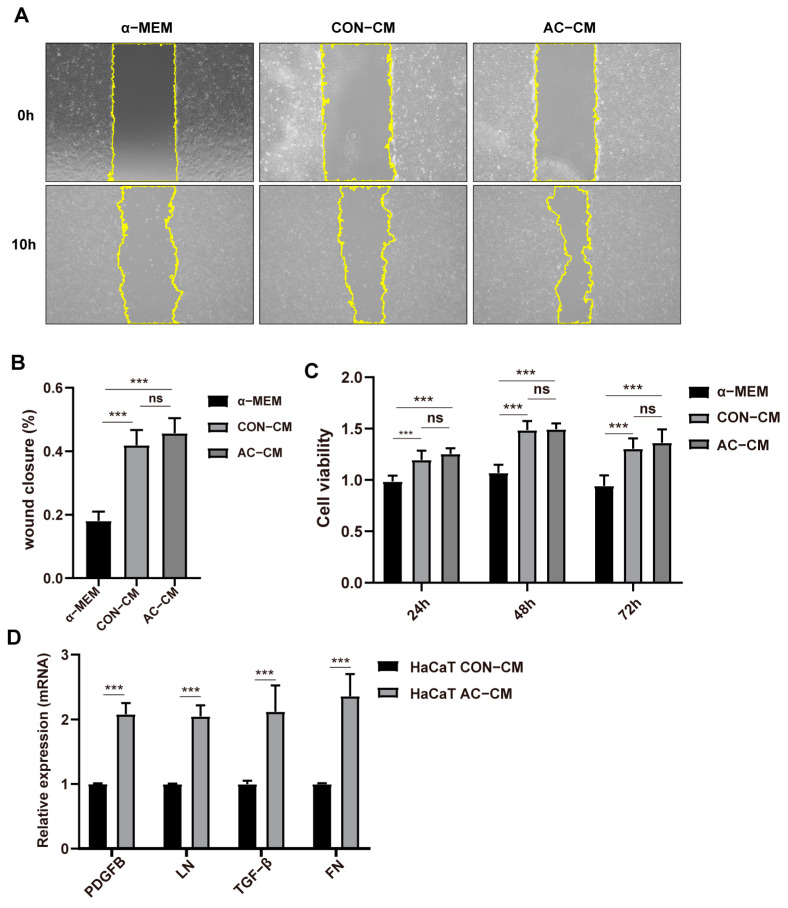
Conditioned medium from AC–ADSCs promotes ECM genes transcription in HaCaT cells. (**A**) Representative scratch test image of HaCaT cells incubated with conditioned medium of CON–ADSCs and AC–ADSCs at 0 and 10 h, and (**B**) Rate of wound area closure after 10 h (%) (*n* = 6, scale bar = 100 μm). (**C**) CCK8 assay was used to evaluate the viability of HaCaT cells incubated with basal medium and conditioned medium of CON–ADSCs and AC–ADSCs (*n* = 6), and the statistical analysis of HaCaT cells viability within 72 h (%). (**D**) RT–qPCR analysis in HaCaT cells to detect ECM genes transcription after incubated with basal medium conditioned medium of CON–ADSCs and AC–ADSCs for 48 h (*n* = 3). Data are shown as mean ± SD. Independent-sample *t* test (two–tailed) was used for statistical comparisons between 2 groups; One–way ANOVA followed by Tukey post hoc test was used for statistical comparisons between multiple groups. **** p* < 0.001.

**Figure 6 ijms-24-05949-f006:**
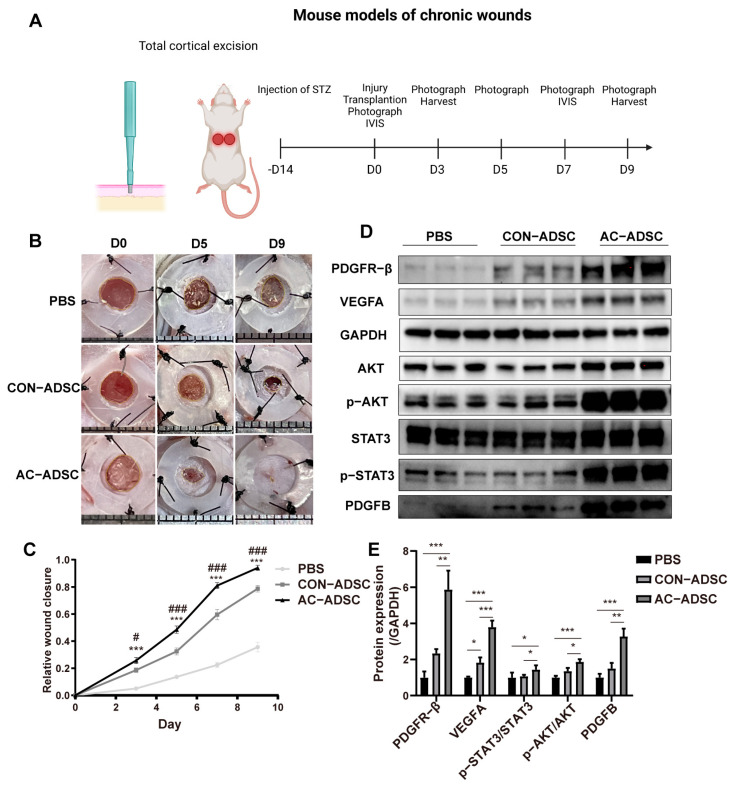
PDGFR–β overexpressed ADSCs accelerated wound healing in diabetic mice. (**A**) Schematic view of the experiments performed on a mouse model of chronic wounds. (**B**) Representative image of excisional wounds in C57BL/6 mice from CON–ADSCs group, AC–ADSCs group and PBS group respectively on D0, D5 and D9. (**C**) Quantitative evaluation of wound closure rates in PBS wounds (*n* = 4), CON–ADSCs wounds (*n* = 10), and AC–ADSCs wounds (*n* = 10). (**D**) Representative images of western blotting and (**E**) quantification analysis of PDGFR–β, VEGFA, P–AKT, and P–STAT3 proteins expression in D9 samples under treatment of PBS, CON–ADSCs and AC–ADSCs (*n* = 3). Data are shown as mean ± SD. Independent–sample *t* test (two–tailed) was used for statistical comparisons between 2 groups; One–way ANOVA followed by Tukey post hoc test was used for statistical comparisons between multiple groups. * *p* < 0.05; ** *p* < 0.01; *** *p* < 0.001, # *p* < 0.05; ### *p* < 0.001.* AC–ADSC versus CON–ADSC, # CON–ADSC versus PBS.

**Figure 7 ijms-24-05949-f007:**
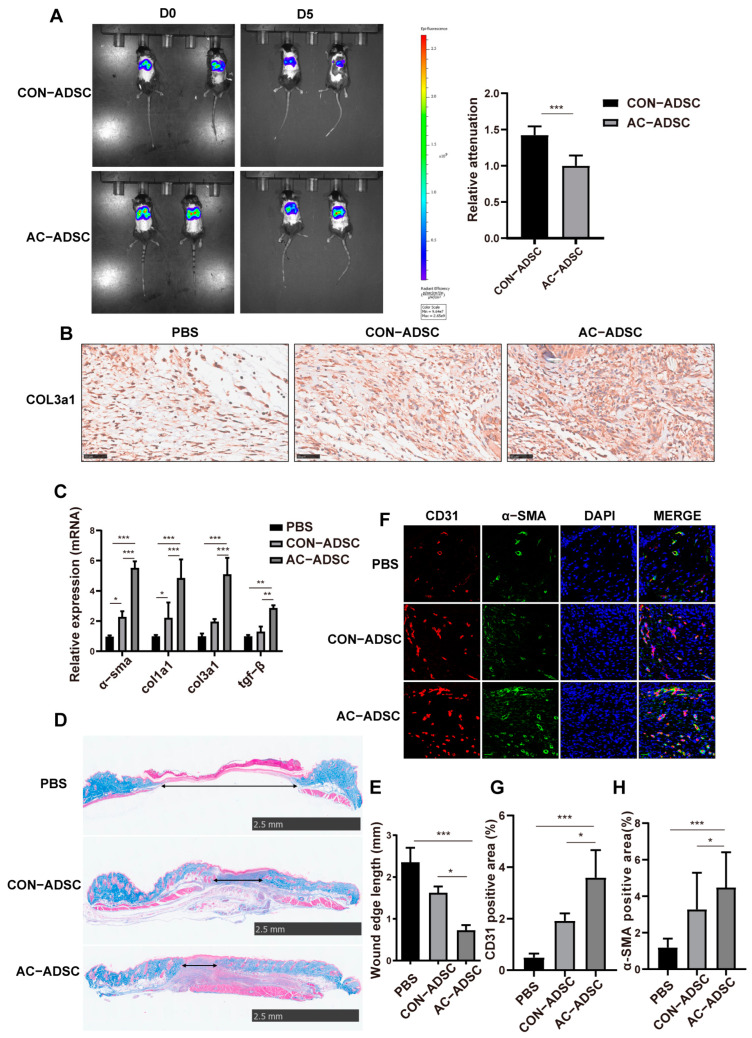
AC–ADSCs accelerate the healing of chronic wounds by enhancing residence at the wound as well as enhancing extracellular matrix remodeling and angiogenesis at the wound. (**A**) Representative image of DIR labeled CON–ADSCs and AC–ADSCs reside around wounds in diabetic mice and quantification analysis of relative attenuation from D0 to D5 (*n* = 5). (**B**) Representative images of COL3a1 immunohistochemistry staining of D3 wounds under treatment of PBS, CON–ADSCs and AC–ADSCs (scale bar = 50 μm). (**C**) RT–qPCR analysis of ECM genes in D3 samples (*n* = 3). (**D**) Histological analysis of wounds at D9 (Masson stain) and (**E**) quantification analysis of wound ledge length (*n* = 3, scale bar = 2.5 mm). (**F**) Representative images of CD31 (red fluorescent signals) and α–SMA (green fluorescent signals) immunofluorescent staining of D9 samples and under treatment of PBS, CON–ADSCs and AC–ADSCs and (**G**,**H**) quantification analysis (*n* = 3, scale bar = 50 μm). Data are shown as mean ± SD. Independent–sample *t* test (two–tailed) was used for statistical comparisons between 2 groups; One–way ANOVA followed by Tukey post hoc test was used for statistical comparisons between multiple groups. ** p* < 0.05; *** p* < 0.01; **** p* < 0.001.

**Figure 8 ijms-24-05949-f008:**
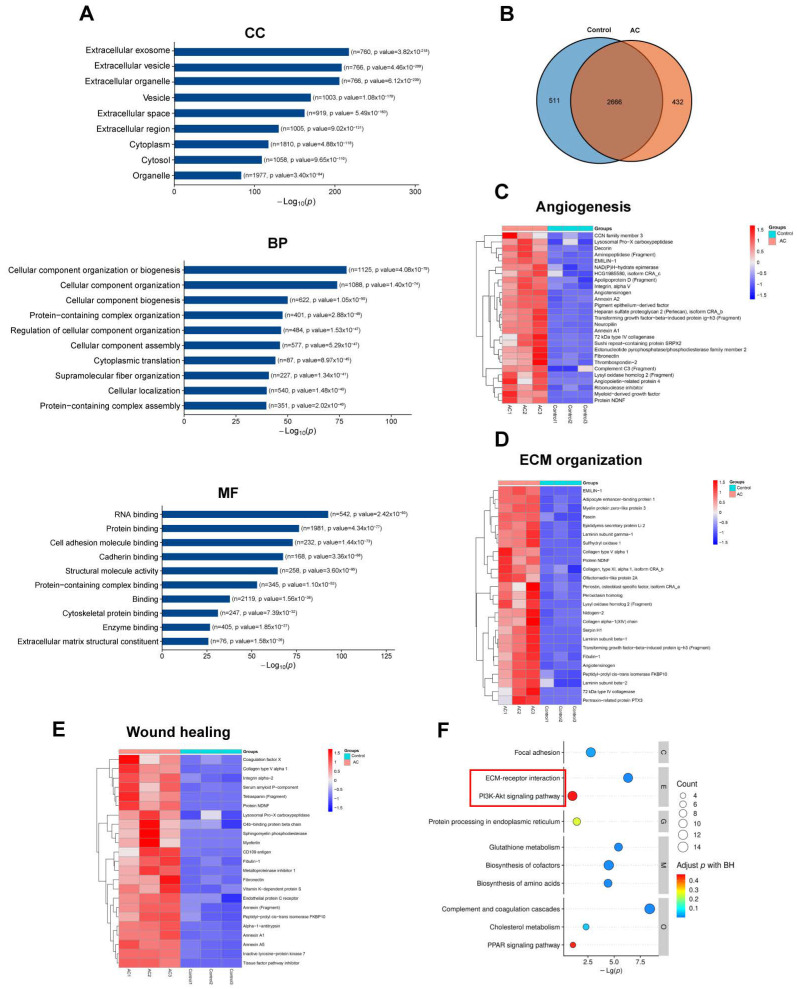
Proteomics of CON–ADSCs CM and AC–ADSCs CM. (**A**) GO analysis of all proteins in ADSCs CM. (**B**) Differential protein Venn diagram of CON–CM and AC–CM. (**C**–**E**) Heat maps of GO enrichment of CON–ADSCs and AC–ADSCs in Angiogenesis, Wound healing and ECM organization. (**F**) KEGG pathway enrichment analysis of upregulated proteins in AC–ADSC CM compared to CON–ADSC CM. Data are shown as mean ± SD. Independent-sample *t* test (two-tailed) was used for statistical comparisons between 2 groups; One-way ANOVA followed by Tukey post hoc test was used for statistical comparisons between multiple groups.

## Data Availability

The proteomics data presented in this study are available in the Appendix A.

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
