# Peer review of "dCas9-Based PDGFR–β Activation ADSCs Accelerate Wound Healing in Diabetic Mice through Angiogenesis and ECM Remodeling"

_ijms, 2023, doi:10.3390/ijms24065949_

Round 1

Reviewer 1 Report

Dear authors,

the manuscript present a very interesting and relevant topic. Chronic wounds are a widespread disease, especially considering that the world average age is rising. In addition, there are many diseases related to the onset of skin ulcers that unfortunately often lead to inauspicious destinies for patients.

Here some of my comments for the manuscript:

Abstract: This acronym used is for PDGF beta receptor, but from the abstract the readrs undertand that the author would indicate PDGF-beta as a growth factor. probabli is only a missing of the receptor word, becaus ethe all manuscript in centered in the pPDGF beta receptor as the acronim used.

Line 42: Probably the authors want to say that MSC promote the wound healing process, and not promote the wound.

Line 46-48: The authors should rewrite the phrase because there is a no consistency respect the previous phrase. The author write that the ADMSCs are aboundant and then write that they are in an insufficient amount.

Line 50: CD104b is PDGF beta receptor, so the authors what they overexpress on their cells? ther is no correspondance with the abstract.

Line 98: The author have to introduce the meaning of an acronym before to use it in the main test.

Line 115: There is no needing to repeat the acronym previously introduced.

Line 136: The animals were fed with a special diet?

Line 179: Probably the paragraph reported will be another paragraph, that will be put before the statistical analysis

Line 201: The authors should introduce in M&M how they differenciate the cells and the staining method used.

Figure 2 caption: D: The authors should indicate with the same acronym respect to the figure. E: The graph indicate the level of PDGFR-beta respect to the actin. F-G: Probably there is a switch in the legend of figures g with f. H-I: Why for this WB the authors used GAPDH instead Actin like others?

Line 235-238: Probably is better to move this part in the M&M section.

Line 248: Were also the AC-ADSCs tested by flow citometry for the typical MSC marker?

Figure 6 captation: D: Did the mice recieve three different injection? I understand only one injection of cells or PBS form M&M. Rewrite the method or correct the legend.

Figure 7 captation: A: Should authours put also the control animal in the imaje to demonstrate that there is no interfirence of PBS with the analisys? B: The author, from the schematic view of the experiment (Fig.6) wrote that the recoveri of the skin was performed only at day 9, but form the M&M wrote that the recovery was perform at day 3 and 9. Probably the schematic experiment need a reviewing.

Line 487: Is not a contrast is a new founding? The previous report, declare the uncontinuous expression of PDGFR-alpha

Line 489: Should authors explain why these two cited works reported the opposit of the presented manuscript? How could the authors hypothesize the the same effects from different approach?

Author Response

Dear Reviewer,

Reviewer 2 Report

The article is a very interesting experimental study on a hot topic: improving wound healing in diabetes. The methodology is clearly described, the results are supported by rich imagistic material.

As a clinician however, I would appeeciate if the authors could add a paragraph in the Discussions section regrading the clinical implications of their findings and new possible therapeutic approaches.

Author Response

Dear reviewer, 

Round 2

Reviewer 1 Report

Thank you for all the responce to my rivision.

Author Response

Thank you very much for your valuable suggestions to promote our manuscript!